# TOWARD EFFECTIVE TOOL-INTEGRATED REASONING VIA SELF-EVOLVED PREFERENCE LEARNING

**Yifei Chen, Guanting Dong, Zhicheng Dou**[*]
Renmin University of China
{zhangboguodong, dou}@ruc.edu.cn

## ABSTRACT

Tool-Integrated Reasoning (TIR) enables large language models (LLMs) to enhance their internal reasoning ability by integrating external tools. However, models with TIR often exhibit suboptimal behaviors, including insufficient tool calls, excessive tool calls, and overthinking after receiving tool call results. How to empower LLMs to perform TIR efficiently and accurately, while stabilizing the reasoning process, remains an open challenge. In this paper, we first analyze the impact of tool calls on model reasoning from the perspective of information entropy. We find that when tool call results are provided, the information entropy of subsequent reasoning content will show a clear trend of change, and the overall information entropy of the reasoning chain will vary depending on the number of tool calls. Based on these observations, we propose Tool-Light, a framework designed to encourage LLMs to perform TIR efficiently and accurately. Our framework consists of dataset construction and multi-stage fine-tuning. For dataset construction, we use the trained model for continuous self-evolved sampling, integrating two methods: vanilla sampling and entropy-guided sampling. At the same time, during the sampling process, we design strict criteria for selecting positive-negative pairs. For the training process, we introduce a two-stage method, which includes a Supervised Fine-Tuning (SFT), and Self-Evolved Direct Preference Optimization (DPO). Test results on 10 datasets reveal the effectiveness of Tool-Light, significantly improving the efficiency and accuracy of the model in completing TIR tasks. Our codes are released at https://github.com/RUC-NLPIR/Tool-Light.

## 1 INTRODUCTION

Recently, large language models (LLMs) have demonstrated strong reasoning abilities after training (Team et al., 2025; Li et al., 2025i; Huang & Chang, 2023; Sun et al., 2023). They excel in tasks such as mathematical reasoning (Shao et al., 2024; Yuan et al., 2023; Li et al., 2024b), logical puzzles (Xie et al., 2025), and code generation (Zhong et al., 2024; Li & Wang, 2025). However, when faced with more challenging tasks (such as deep information retrieval and precise computation (Min et al., 2024; Qin et al., 2024)), those trained LLMs often struggle when relying solely on their internal reasoning capabilities. To solve those problems, Tool-Integrated Reasoning (TIR) (Paranjape et al., 2023; Gou et al., 2024; Li et al., 2025b; Wang et al., 2025d) has emerged. Compared to general reasoning models, TIR models can autonomously utilize external tools during the reasoning process to compensate for the deficiencies in their internal knowledge or abilities (Chen et al., 2025a; Song et al., 2025; Li et al., 2024a; 2025b;f; Dong et al., 2025c;b).

While introducing external tools enables models to enhance performance on reasoning tasks, there still exist challenges, such as unreasonable tool calls or overthinking. For example, during the TIR process, the model often exhibits suboptimal tool call patterns, such as excessive or insufficient tool calls (Wu et al., 2025b; Bai et al., 2025). Additionally, when low-quality tool call results are provided, models may experience overthinking or even analysis paralysis (Zheng et al., 2025; Sui et al., 2025; Chen et al., 2025b). We collectively refer to them as incorrect tool calls. Recent studies have focused on optimizing tool calls with the help of reinforcement learning (RL). These

---

[*]Corresponding author.

studies include scenarios of using a single tool and scenarios of using multiple tools (Bai et al., 2025; Feng et al., 2025a; Wu et al., 2025b; Wang et al., 2025e; Dong et al., 2025a; Wang et al., 2025a). Nevertheless, existing studies typically focus on the issue of tool-overuse, neglecting both tool underuse and the impact of tool call results on subsequent reasoning processes (Qian et al., 2025). We consider that these works do not comprehensively address incorrect tool call issues.

At present, many works have analyzed reasoning tasks from the perspective of information entropy (Cui et al., 2025; Wang et al., 2025c). They find that the high-entropy part of the reasoning chain often determines reasoning direction. Inspired by these works, we conduct preliminary experiments and analyze the TIR task's information entropy. We find that after receiving tool call results, the information entropy of the model's subsequent outputs will fluctuate. We also observe that when multiple correct reasoning paths exist, the path with fewer tool calls tends to have a lower overall entropy distribution compared to those with more. Based on the limitations of existing work and the TIR task's entropy characteristics, we attempt to explore the efficiency of the TIR task from the following two perspectives:

- **From the training perspective:** Can we improve the effectiveness of tool call in TIR tasks from the perspective of post-training (including the algorithm side and the data side)?
- **From the inference perspective:** How can we apply the entropy distribution characteristics of the model's TIR process to guide data sampling?

In this paper, we propose Tool-Light, a framework that optimizes the reasoning capability and tool call effectiveness of TIR models from the data construction and algorithmic perspectives. From the data construction perspective, we introduce an innovative entropy-guided sampling method. This approach first generates a main chain, then branches from the highest-entropy positions to create multiple paths. We then apply sampling criteria to derive high-quality positive-negative pairs. To ensure data diversity and balance, we integrate this approach with direct sampling, producing a hybrid method that sustains performance while improving tool-call effectiveness. From the algorithmic perspective, we propose the "Two-stage TIR Training" pipeline. First, we perform supervised fine-tuning (SFT) on the original model, and then conduct self-evolved DPO training. In self-evolved DPO stage, we divide the training into two parts: Pre-Aligned DPO Training and Self-Evolved DPO Alignment. The former is used to improve the model's reasoning ability while reducing the redundant tool calls. The latter alternates sampling and training process, enabling the model to learn necessary tool call. At the same time, the difficulty of the training data and the model's capability can be adapted, thereby maintaining its original reasoning ability.

To comprehensively evaluate the capabilities of Tool-Light, we use ten challenging reasoning tasks, including knowledge-intensive and mathematical-reasoning tasks. Tool-Light not only ensures stable reasoning performance but also significantly improves the efficiency and accuracy of tool calls. In summary, our contributions are as follows:

- We explore and analyze the TIR paradigm from the perspective of information entropy, demonstrating the connection between TIR effectiveness and entropy change.
- We propose an innovative entropy-guided sampling strategy, which is combined with a two-stage training method incorporating a self-evolution mechanism, thereby enhancing the effectiveness of the TIR process.
- Experiment results across 10 challenging reasoning datasets prove the effectiveness of Tool-Light. Further quantitative analyses offer practical guidance for efficient tool-integrated reasoning.

## 2 RELATED WORK

**Effective Tool-integrated Reasoning.** Tool-Integrated Reasoning (TIR) refers to guiding models to leverage external tools during reasoning (Yang et al., 2025). Teaching models to use tools correctly and rationally is a core challenge in TIR (Su et al., 2025a; Fang et al., 2025; Li et al., 2025d;c; Tao et al., 2025; Li et al., 2025j; Qiao et al., 2025; Qin et al., 2023; Schick et al., 2023; Lu et al., 2023; Dong et al., 2026). Currently, fine-tuning is the main method to guide the model in efficiently completing the TIR task (Li et al., 2025e). For example, IKEA (Huang et al., 2025) and SMART (Qian et al., 2025) design training methods based on metacognitive theory (Kontostavlou

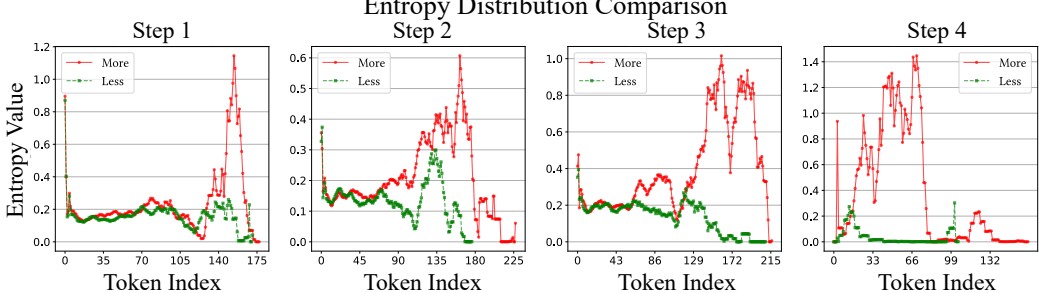

Figure 1: The entropy distribution of tool-integrated reasoning inference. "Step" refers to the reasoning steps—from the model starting inference or receiving the result of the previous tool call, up until the next tool call is triggered or the final answer is reached—including the process of triggering the tool call (but not including the result of the tool call). Step 1-4 thus represents the process of intermediate reasoning in the first, second, third, and fourth segments. "Token Index" represents the position of a token within that step. For example, $index_i$ represents the subscript index of the $i$-th output token in a certain step.

& Drigas, 2021), focusing on the model's knowledge boundaries. Self-DC framework (Wang et al., 2025b) controls the model's self-behavior by leveraging internal signals. ToolPrefer-LLaMA (Chen et al., 2025c) uses the DPO algorithm to update LLM strategies. Additionally, some works efficiently complete the TIR task by optimizing the reinforcement learning training process. For instance, Search Wisely (Wu et al., 2025b) and OTC (Wang et al., 2025a) carefully designed reward functions during training, while CoRT (Li et al., 2025a) meticulously optimized the training process. However, most existing work focuses solely on reducing the excessive tool calls and does not explore the accuracy of tool calls and the reasoning process within TIR. We consider that the effectiveness of TIR includes not only reducing redundant tool calls, but also invoking necessary tools. Moreover, it also involves minimizing overthinking during the reasoning process and avoiding analysis paralysis. In this paper, we do not limit our discussion to reducing redundant tool calls, but explore the effectiveness of TIR in a broader range of scenarios.

**Self-Evolved Methods in LLMs.** Due to the variability and complexity of world knowledge, models need to continuously evolve to better adapt to the real-world tasks (Wu et al., 2025a; Qiao et al., 2025; Dong et al., 2025c; Li et al., 2025d; Tao et al., 2025). The self-evolution strategy refers to a paradigm for dynamically improving model output performance through continuous learning and adaptation (ang Gao et al., 2025). It can be divided into two methods: parameter-updating and in-context learning. The former involves having the model autonomously generate training data to fine-tune weights, or using the experience from interacting with the environment to drive model training(Zhou et al., 2025; Simonds et al., 2025; Zeng et al., 2025; Su et al., 2025b). The latter achieves model self-evolution through the optimization of prompt words or memory mechanisms(Chhikara et al., 2025; Yüksekgönül et al., 2024). Referring to existing research, the core of Tool-Light lies in enabling the model to learn from data generated by itself(Li et al., 2025g). At the same time, it continuously generates better training data to perform parameter-updating self-evolution.

## 3 PRELIMINARIES

**(1) Problem Definition.** Multi-TIR aims to enable the model to autonomously call multiple tools, thus completing different complex tasks. Specifically, given the input instruction $\mathcal{I}$ and the model $\theta$, the final output answer $y$ of multi-TIR can be expressed in the following form:

$$y \sim P_\theta(y \mid R_N) \cdot \prod_{i=1}^{N} P_\theta(R_i \mid T_{C<i}, R_{<i}, \mathcal{I}), \qquad (1)$$

where $i$ represents the reasoning step, $N$ represents the total number of reasoning steps, $R_{<i}, T_{C<i}$ and $R_i$ represent the reasoning content before the $i$-th step, the tool call result before the $i$-th step,

and the reasoning content of the $i$-th step, respectively. For the tool $T$, it may contain a variety of different tools in multi-TIR tasks.

**(2) Pre-Experiment.** In this section, we explore the relationship between the TIR path and the distribution of information entropy. First, we present the formula for calculating the information entropy distribution:

$$H(i) = -\sum_{j=1}^{N} P(y_{ji}|y_{<i}) log P(y_{ji}|y_{<i}), \tag{2}$$

where $N$ represents the vocabulary length at the $i$-th position, $y_{<i}$ and $y_{ji}$ represent the sequence before position $i$ and the $j$-th token at position $i$, respectively. We use the Search-R1 (Jin et al., 2025a) model to perform TIR on multiple QA datasets. For each sample, we rollout ten chains and classify them into two groups based on the tool calls: more tool calls (**more**) and fewer tool calls (**less**). We record the average entropy distribution for each step. Our results can be found in Figure 1.

From the results of this experiment, we can draw some conclusions:

- When the model receives a tool call result, its output information entropy initially rises, then fluctuates, and finally drops sharply before the next tool call arrives.
- For the same sample, low-entropy chains tend to involve fewer tool calls, and as reasoning progresses, tool calls' difference between high and low entropy chains becomes increasingly evident.

These conclusions guide our subsequent data sampling.

**(3) Tool Design** Inspired by the existing work, we choose the two most mainstream tools, the code interpreter tool and the search tool, for exploration (Dong et al., 2025a; Wang et al., 2025a; Li et al., 2025h; Jin et al., 2025a). The code interpreter tool receives a piece of code, compiles it, and returns the compilation result or error message. The search tool receives a query, and then retrieves the content most relevant to the query from the local knowledge base or web page and returns it.

## 4 METHOD

**Overview.** We propose Tool-Light, a multi-stage training pipeline aiming to improve the effectiveness of model tool calls. As shown in Figures 2 and 3, Tool-Light consists of two key components: **(1) Dataset construction**, which includes carefully designed sampling strategies to screen out training data. **(2) Two-stage TIR training paradigm**, which trains the model successively with SFT and self-evolved DPO training. In the self-evolved DPO training stage, we design pre-aligned DPO training and self-evolved DPO alignment stages to gradually improve the model's capabilities.

### 4.1 DATASET CONSTRUCTION

In this section, a detailed introduction to the data construction and sampling strategies is provided.

**Source Data Construction.** In this section, we first obtain the source data for subsequent sampling. To improve the model's ability to execute TIR tasks, we adopt the SFT data from existing work, defined as $D_{sft}$ (Dong et al., 2025a). We use $D_{sft}$ to train the instruct model, resulting in the model $M_{sft}$. Then we use the instruct model to perform inference on $D_{sft}$. During this process, we do not offer the model external tools. After that, we only retain the data where the model's inference answers are incorrect, resulting in $D_{source}$. This process can be modeled as:

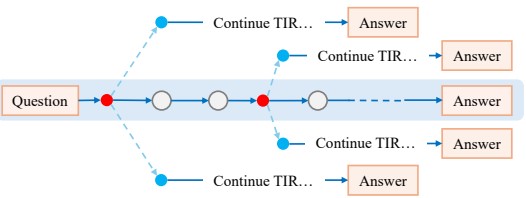

Figure 2: The overall process of entropy-guided sampling. Gray and red positions represent tool calls and the branching positions, respectively.

$$D_{source} = D_{sft}[y_{label} \neq y_{inference}], \tag{3}$$

where $y_{label}$ and $y_{inference}$ are the golden answer and the result obtained from direct inference on that sample, respectively. $y_{label} \neq y_{inference}$ means that we only retain data where the model's inference

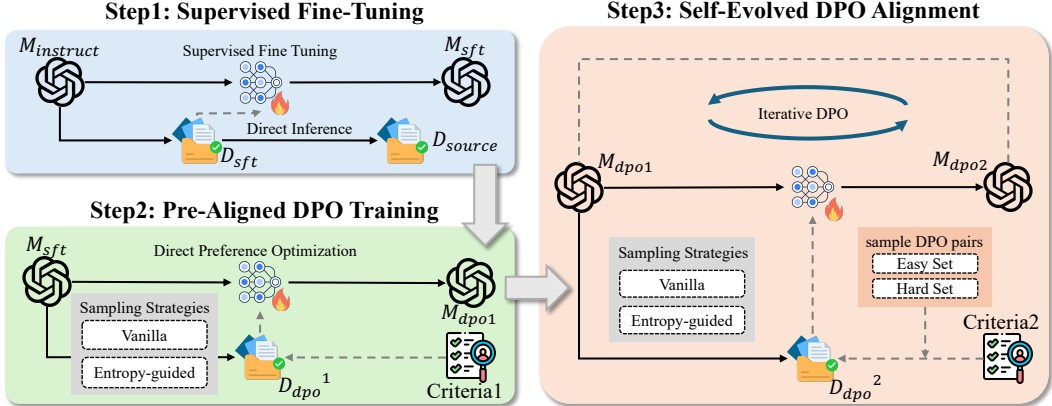

Figure 3: The overall structure of Tool-Light's training pipeline. Among them, the Self-Evolved DPO Alignment stage will conduct multiple rounds of training.

result is incorrect. We consider these data to be more challenging, and training on them can help improve the model's reasoning capabilities.

**Sampling Strategy Design.** After obtaining $D_{\text{source}}$, we use $M_{\text{sft}}$ to perform TIR inference on it. For each question, we sample multiple TIR paths. The final set of all inference paths is denoted as $D_{\text{dpo}}$. For this step, we carefully design two sampling strategies, and mix the data from both strategies to construct high-quality training data. The specific strategies are described as follows:

**(1) Vanilla TIR Sampling.** We guide the model to generate multiple paths for each sample. This helps us construct positive-negative pairs. This process can be expressed as follows:

$$D_{\text{dpo}}^1 = \{y | y = M_{\text{sft}}\left[I(q)\right], q \sim D_{\text{source}}\}, \tag{4}$$

where $D_{\text{dpo}}^1$ represents the sampling results generated under vanilla sampling, $I$, $q$ and $y$ represent the prompt used, each question in $D_{\text{source}}$, and the output result from $M_{\text{sft}}$, respectively.

**(2) Entropy-Guided Sampling.** Vanilla sampling strategy incurs high inference costs and inference uncertainty. In this section, as shown in Figure 2, we propose an entropy-guided sampling strategy.

- **Entropy Distribution Calculation.** We begin by generating a primary reasoning chain, denoted as $C_{\text{main}}$. Each segment of the chain is considered a step. According to existing research (Liao et al., 2025), the initial inference step has a significant impact on the subsequent outputs. Therefore, for each step, we compute the entropy distribution at each token position in the first 10, 20, 30, 40, and 50 tokens. The average information entropy distribution for a subsequence is:

$$H_{\text{avg}}(i) = \frac{1}{i}\sum_{j=1}^{i} H(j), \tag{5}$$

where $H_{\text{avg}}(i)$ represents top-$i$ tokens' average entropy distribution. We retain the maximum $H_{\text{avg}}(i)$ and its corresponding sequence length $i$ at each step.

- **Branch Sampling Execution.** We pick the top-k steps with the highest entropy, along with their sampling positions. At each position, we guide the model to continue generating several branches. Since those subsequences have higher information entropy, based on previous research (Wang et al., 2025c; Cui et al., 2025; Cheng et al., 2025), it is more likely to produce branch outputs with higher diversity. This process can be modeled as follows:

$$D_{\text{dpo}}^2 = \{y | y_{>i} = M_{\text{sft}}[I(q) \oplus y_{<i}], q \sim D_{\text{source}}\}, \tag{6}$$

where $D_{\text{dpo}}^2$ and $y_{>i}$ represent the dataset guided by entropy sampling and the subsequent sampling output, respectively. $I(q) \oplus y_{<i}$ represents concatenating the prompt and the existing sequence together as the input. $y$ represents concatenating $y_{>i}$ and $y_{<i}$ together. Assuming $m$

rollout times and an average sequence length of $n$, this method, in the ideal scenario, reduces computational complexity from $O(mn)$ to $O(n \log m)$. We collect all the main chains and branch chains, and construct the set of outputs based on tree-structured sampling.

Finally, $D_{\text{dpo}}^1$ and $D_{\text{dpo}}^2$ are mixed in a certain ratio to obtain the final dataset $D_{\text{dpo}}{}^1$ for the positive-negative pairs construction.

## 4.2 TWO-STAGE TIR TRAINING PARADIGM

Based on existing research (Li et al., 2025g; Dong et al., 2025a; Song et al., 2025), we propose a two-stage self-evolved training pipeline to gradually boost the effectiveness and stability of the model's TIR process. The specific pipeline is shown in Figure 3.

**Supervised Fine-Tuning.** First, we obtain the SFT model $M_{sft}$ with the same setup as in Tool-Star (Dong et al., 2025a). The SFT's loss function is: $\mathcal{L}_{\text{SFT}}(\theta) = -\sum_{(x,y) \in D} \log P_\theta(y|x)$. As shown in the first step of Figure 3, this step aims to help the model quickly acquire the ability to complete TIR tasks.

**Self-Evolved DPO.** Inspired by existing works (Li et al., 2025g; Wu et al., 2023; Tao et al., 2024), we further fine-tune $M_{\text{sft}}$ through a self-evolved DPO strategy, using a loss function as follows:

$$\mathcal{L}_{\text{DPO}}(\pi_\theta; \pi_{\text{ref}}) = -\mathbb{E}_{(x,y_w,y_l) \sim D} \left[ \log \sigma \left( \beta \log \frac{\pi_\theta(y_w|x)}{\pi_{\text{ref}}(y_w|x)} - \beta \log \frac{\pi_\theta(y_l|x)}{\pi_{\text{ref}}(y_l|x)} \right) \right], \quad (7)$$

where $\pi_{\text{ref}}$ is the original strategy model, and $\beta$ is the hyperparameter and $\sigma$ is the sigmoid function. The primary objective of DPO is to maximize the relative probability of the model generating $y_w$ compared to $y_l$. This step aims to endow the model with the capability to finish TIR tasks accurately and efficiently. First, we establish a specialized data sampling criterion $\text{Cri}_1$. It is used to obtain training data $D_{\text{dpo}}{}^1$ from $M_{\text{sft}}$ using both vanilla and entropy-guided strategies. We then train $M_{\text{sft}}$ on $D_{\text{dpo}}{}^1$ to yield $M_{\text{dpo1}}$. Subsequently, a new criterion $\text{Cri}_2$ is defined, guiding $M_{\text{dpo1}}$ to sample the next iteration of training data $D_{\text{dpo}}{}^2$. We train $M_{\text{dpo1}}$ on this dataset and get $M_{\text{dpo2}}$. After that, we update $M_{\text{dpo1}} \leftarrow M_{\text{dpo2}}$ and iteratively repeat the sampling and training steps until performance converges. Details on these two criteria follow in the subsequent sections.

*(1) Pre-Aligned DPO Training.* In each sample, we first categorize the correct or incorrect trajectories based on the F1 score. Only trajectories with 1 F1 score are considered correct, and only trajectories with 0 F1 score are considered incorrect. Then, we divide the samples into a hard set and an easy set. After that, referring to the conclusions in Section 3, we design criteria for sampling positive and negative pairs in the entropy-guided strategy:

- **Hard set:** Samples with $\leq 50\%$ correct trajectories.
- **Easy Set:** Samples with $\geq 50\%$ correct trajectories.
- **Positive Example:** Correct trajectory with the minimal number of tool calls and lowest entropy. If no correct trajectory exists, the corresponding SFT trajectory from $D_{\text{source}}$ is used.
- **Negative Example:** An incorrect trajectory with more tool calls than the positive example.

For vanilla sampling criteria, we choose samples with $\leq 40\%$ correct trajectories as the hard set, and those with $\geq 70\%$ correct trajectories as the easy set. For each set, the shortest correct trajectory is the positive example, and the incorrect trajectory longer than the positive one is the negative example.

We set the hard set and easy set's data ratio as 2:1. The goal of training in this round is to teach the model to reduce unnecessary tool calls while avoiding excessive reasoning.

*(2) Self-Evolved DPO Alignment.* We use $M_{\text{dpo1}}$ to resample from $D_{\text{dpo}}{}^1$. At the same time, we introduce the second criterion to filter out $D_{dpo}{}^2$. In $\text{Cri}_2$, samples whose number of correct trajectories is less than half of the incorrect ones are classified into the hard set, while the criteria for the easy set remain unchanged. For each sample generated by $M_{dpo1}$, if it is classified into the easy set, it means that after pre-aligned DPO training, $M_{dpo1}$ has mastered the knowledge to solve this

problem. If it is classified into the hard set, it means that this problem is still difficult for $M_{dpo1}$. For these two different situations, we have also designed the sampling criteria for positive and negative examples:

- **Negative Example in Easy Set:** Incorrect trajectory with the most tool calls.
- **Positive Example in Easy Set:** Correct trajectory with fewer tool calls and lowest entropy.
- **Positive Example in Hard Set:** Correct trajectory with the longest reasoning chain.
- **Negative Example in Hard Set:** Incorrect trajectory with the shortest reasoning chain.

Based on preliminary experiments, low-entropy outputs' tool calls are often fewer. Therefore, we also incorporate the consideration of low entropy into positive examples' selection. We use the sampled $D_{\text{dpo}}{}^2$ to continue training $M_{dpo1}$, and continuously sample with the trained model. Self-evolved DPO alignment will continue for several loops. This step is intended to equip the model with the ability to make necessary tool calls while maintaining its original efficient TIR capability. After several rounds of training, we obtain the final model $M_{\text{dpo2}}$.

## 5 EXPERIMENTS

### 5.1 EXPERIMENTAL SETUP

**Dataset.** We carefully select 10 datasets and categorize them into two types: **(1) mathematical-reasoning tasks**, including AIME24, AIME25 [1], AMC23 [2], MATH (Hendrycks et al., 2021), MATH500 (Lightman et al., 2024), and GSM8K (Cobbe et al., 2021), and **(2) knowledge-intensive tasks**, including HotpotQA (Yang et al., 2018), 2WikiMultiHopQA (Ho et al., 2020), MuSiQue (Trivedi et al., 2022), and Bamboogle (Press et al., 2023).

**Metrics.** For mathematical-reasoning tasks, we adopt the LLM-as-Judge approach (Gu et al., 2024), utilizing the Qwen2.5-72B-Instruct model to evaluate answer correctness. For knowledge-intensive tasks, we use F1 score to evaluate answers. Additionally, we introduce two metrics: **Efficiency** and **Necessity**. **(1) Efficiency** is defined as $\text{Effi} = \frac{1}{n}\sum_{i=1}^{n}\frac{M_i}{T_i}$, where $n$ is the total number of samples, $M_i$ the performance on the $i$-th sample, and $T_i$ the number of tool calls. It captures the model's tendency to overuse tools. **(2) Necessity** is defined as $\text{Nece} = \mathcal{M}\left(\frac{1}{n}\sum_{i=1}^{n}(N_{in}^i - N_{co}^i)\right)$, where $N_{in}^i$ is the count of methods for the $i$-th sample that make more tool calls than the current method but yield incorrect answers, $N_{co}^i$ is the count of methods with fewer tool calls yet correct answers, and $\mathcal{M}$ denotes Min-Max Scaling. It reflects the model's tendency to underuse tools.

**Baselines.** We classify baseline methods into two types: **(1) Single-Tool-Integrated Reasoning**, including Search-o1 (Li et al., 2025f), Search-R1 (Jin et al., 2025a), DotaMath (Li et al., 2024a), ToRL (Li et al., 2025h), and ReTool (Feng et al., 2025b), and **(2) Multi-Tool-Integrated Reasoning**, including prompt engineering, ReCall (Chen et al., 2025a), and ToolStar (Dong et al., 2025a).[3]

### 5.2 MAIN RESULTS

Our main results are presented in Table 1, and more details can be found in Appendix 3. We can draw the following insights:

**(1) Proper use of external tools can be of great help to model reasoning.** For example, when external tools are introduced, the prompting-based method on the Qwen model has significant advantages over the direct inference method in mathematical reasoning tasks. However, for knowledge-intensive tasks, the effect of introducing external tools is reduced (only improved on MuSiQue dataset). This also indicates that untrained models cannot use external tools well.

**(2) While training enhances the model's TIR capabilities, single-tool training hinders the development of generalization abilities.** Compared to using prompt engineering methods, fine-tuning

---

[1] https://huggingface.co/datasets/AI-MO/aimo-validation-aime

[2] https://huggingface.co/datasets/zwhe99/amc23

[3] More implementation details can be found in Appendix B.

Table 1: Results on 10 reasoning tasks, with top two results highlighted in **bold** and underlined. Unless noted, Qwen2.5-7B-Instruct is used as the backbone. For ToRL and ReTool, their RL models generate training data inferences for SFT, which serves as the baseline. Abbreviation: 2Wiki. (2WikiMultiHopQA). HQA. (HotpotQA). MSQ. (MuSiQue). Bamb. (Bamboogle).

| Method | Mathematical-Reasoning Tasks | | | | | | Knowledge-Intensive Tasks | | | | Avg. |
|---|---|---|---|---|---|---|---|---|---|---|---|
| | AIME24 | AIME25 | AMC23 | MATH | MATH500 | GSM8K | HQA | 2Wiki. | MSQ | Bamb | |
| *Direct Inference* | | | | | | | | | | | |
| Qwen2.5-7B-Instruct | 0.0 | 6.7 | 30.0 | 68.6 | 57.2 | 71.4 | 26.1 | 25.6 | 7.9 | 36.5 | 33.0 |
| Llama3.1-8B-Instruct | 0.0 | 3.3 | 15.0 | 52.8 | 33.4 | 75.0 | 16.2 | 13.7 | 7.4 | 23.2 | 24.0 |
| *Single-TIR Methods* | | | | | | | | | | | |
| Search-o1 | 6.7 | 10.0 | 37.5 | 73.6 | 61.8 | 80.2 | 41.1 | 35.4 | 13.2 | 39.8 | 39.9 |
| Search-R1 | 16.7 | 6.7 | 45.0 | 81.2 | 63.8 | 82.4 | 48.7 | 40.0 | 24.1 | 47.4 | 45.6 |
| DotaMath | 16.7 | 10.0 | 50.0 | 74.6 | 62.2 | 82.6 | 26.2 | 21.7 | 6.5 | 28.6 | 37.9 |
| ToRL | 30.0 | 26.7 | 67.5 | 87.0 | **80.2** | 89.2 | 41.3 | 35.4 | 9.5 | 36.9 | 50.4 |
| ReTool | 23.3 | **30.0** | 62.5 | 84.8 | 78.4 | 86.2 | 31.5 | 29.0 | 11.1 | 35.8 | 47.3 |
| *Multi-TIR Methods* | | | | | | | | | | | |
| Prompting-Based | 6.7 | 13.3 | 47.5 | 73.8 | 62.2 | 69.4 | 21.1 | 23.8 | 9.9 | 25.5 | 35.3 |
| ReCall | 3.3 | 6.7 | 27.5 | 73.2 | 54.6 | 79.8 | 51.9 | 54.0 | **25.0** | 55.5 | 43.2 |
| Tool-Star | 30.0 | 26.7 | 65.0 | 85.6 | 77.2 | 89.4 | 54.7 | 55.7 | 22.8 | **58.8** | 56.6 |
| *Ours* | | | | | | | | | | | |
| Tool-Light (Llama) | 10.0 | 6.7 | 30.0 | 59.4 | 56.8 | 76.6 | 41.3 | 33.5 | 12.2 | 41.3 | 36.8 |
| Tool-Light (Qwen) | **33.3** | 23.3 | **67.5** | **87.4** | 79.0 | **92.0** | **57.7** | **56.1** | **25.0** | 58.7 | **58.0** |

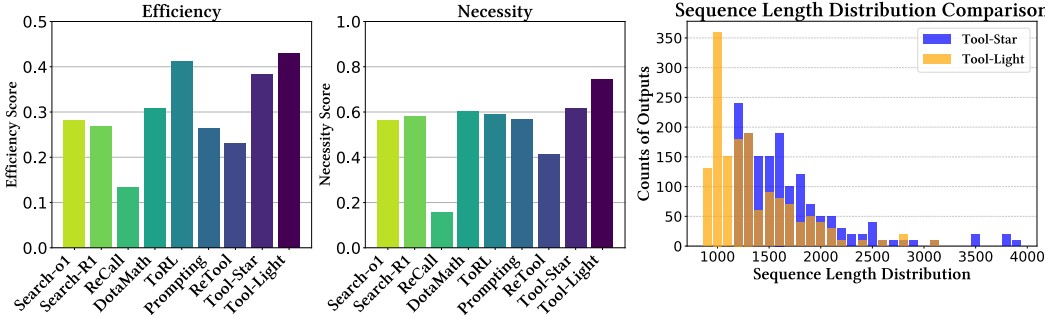

Figure 4: Efficiency, Necessity, and Sequence Length differences between Tool Light and baselines.

can effectively teach the model to use tools. For example, Search-R1 outperforms Search-o1 by 5.7 on average, while Tool-Star outperforms prompting-based methods by 21.3 on average. It is worth noting that models trained for single-TIR tend to have poor generalization performance. For example, Search-R1 performs well on knowledge-intensive tasks but poorly on mathematical-reasoning tasks, while ToRL shows the opposite pattern. The Tool-Star and Tool-Light frameworks perform well on both types of tasks, highlighting the generalization advantages of multi-TIR training.

**(3) Multi-round DPO training allows for further enhancement of the model's performance.** Compared with the simple SFT method, Tool-Light framework additionally introduces two operations: Pre-Aligned DPO Training and Self-Evolved DPO Alignment. This leads to an overall improvement in the model's performance. For example, in mathematical-reasoning tasks, Tool-Light achieves optimal performance on four datasets, and ranks among the top two on all datasets for Knowledge-Intensive tasks. Notably, by solely adopting the DPO method, we outperform most baselines that are trained using GRPO when it comes to average performance. This fully demonstrates the effectiveness of our training pipeline design.

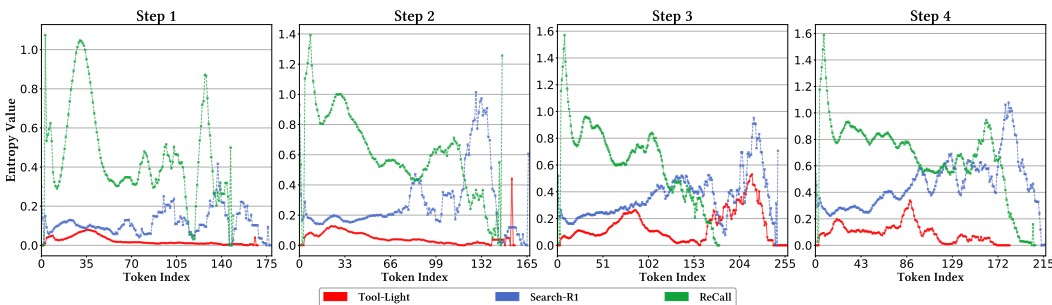

Figure 5: Output sequences' entropy distribution under different methods.

## 5.3 QUANTITATIVE ANALYSIS

**Tool-Use Effectiveness Analysis**   In this section, we analyze the differences in TIR effectiveness between Tool-Light and other baseline methods. As shown in Figure 4, Tool-Light achieves the best performance in Efficiency and Necessity metrics, which indicates the superiority of our designed training process. Regarding the output sequence length metric, Tool-Light has a shorter length distribution than Tool-Star, and outperforms Tool-Star in both result correctness and reasoning effectiveness. It shows that our training method can effectively reduce the phenomenon of overthinking. Detailed cases are list in Appendix D.

**Entropy Distribution Analysis**   As described in Section 3, in this section, we explore the impact of introducing the entropy-guided sampling strategy on final output results' information entropy distribution. The results are shown in Figure 5. The existing TIR models tend to produce output sequences with relatively high entropy. In contrast, Tool-Light results in lower entropy distributions for its output sequences. This might be due to the consideration of information entropy during sampling. Our training method introduces the learning of low-entropy path outputs. Since high-entropy tokens can sometimes trigger excessive reasoning in the model, our method effectively mitigates the phenomenon of overthinking. This also aligns with the results in Figure 4. [4]

**Impact of DPO Loop Number**   In this section, we investigate how the number of training loops in the self-evolved DPO alignment stage influences the model's performance. For this stage, we increase the loop count from 1 to 5 and record the Performance, Efficiency, and Necessity metrics. The results are shown in Table 2's 1-5 rows. As the number of training loop increases, all metrics reach optimal values in the second training round and then decline. This may be because the self-evolved method is initially able to obtain sufficient positive-negative pairs. However, as the evolution proceeds, there are fewer sample pairs beneficial for training, and the model may overfit to the distribution of the training set, leading to lower performance on the test set.

Table 2: Ablation experiment results for various aspects: *1/1 strategy ratio* indicates a 1:1 data ratio for two sampling strategies, *p-r.* and *n-r.* denote random positive and negative example selection.

| Method | Performance | Efficiency | Necessity |
|---|---|---|---|
| Tool-Light (*2 loop*) | 58.0 | 0.44 | 0.75 |
| *Ablation for self-evolved Loops* | | | |
| w. 1 loop | 57.9 (-0.1) | 0.42 (-0.02) | 0.71 (-0.04) |
| w. 3 loop | 56.1 (-1.9) | 0.39 (-0.05) | 0.73 (-0.02) |
| w. 4 loop | 56.4 (-1.6) | 0.37 (-0.07) | 0.71 (-0.04) |
| w. 5 loop | 54.1 (-3.9) | 0.36 (-0.08) | 0.72 (-0.03) |
| *Ablation for Sampling Criteria* | | | |
| w. 1/1 data ratio | 56.9 (-1.1) | 0.44 | 0.76 (+0.01) |
| w. p-r. | 53.6 (-4.4) | 0.42 (-0.02) | 0.63 (-0.12) |
| w. n-r. | 53.9 (-4.1) | 0.41 (-0.03) | 0.74 (-0.01) |

**Impact of Sampling Criteria**   In this section, we explore the impact of sampling criteria on the training effect. We modify the original criteria from the following aspects: the data ratio of two strategies is 1:1, positive examples are

---

[4]We also explore the impact of different data ratios on the final training results during sampling. The experimental results and analyses are provided in Appendix C.

randomly selected, and negative examples are randomly selected. The results are shown in rows 6 - 8 of Table 2. We can find that the different data ratio has the least impact on the training effect, while the change in the positive-negative example selection criteria has a greater impact on the results. This indicates that for the DPO training in Tool-Light, it is very important to fully distinguish between positive and negative examples.

## 6 CONCLUSION

In this paper, we propose Tool-Light, a framework empowering models to efficiently and accurately complete TIR tasks. We first analyze the TIR process from the perspective of information entropy. Based on this analysis, we introduce Tool-Light, a framework improving model's TIR capabilities from both data construction and training aspects. Within this framework, we design a new entropy-guided sampling strategy. Based on this method, we construct a two-stage TIR training paradigm. The trained model achieves excellent results in terms of overall performance, tool calls effectiveness, and simplicity of the inference process. We confirm that our work can provide more assistance for future research on TIR.

## 7 ACKNOWLEDGEMENT

This work was supported by the National Natural Science Foundation of China No.62272467. The work was partially done at the Engineering Research Center of Next-Generation Intelligent Search and Recommendation, MOE.

## 8 ETHICS STATEMENT

All authors of this work have strictly adhered to the ICLR Code of Ethics. It does not involve any research with human subjects, practices to data set releases, potentially harmful insights, methodologies and applications, potential conflicts of interest and sponsorship, discrimination/bias/fairness concerns, privacy and security issues, legal compliance, and research integrity issues.

## 9 REPRODUCIBILITY STATEMENT

To ensure the reproducibility of Tool-Light's experiments, we detail all the experimental settings in Appendix B. Among them, the data ratio based on the difficulty level of data sampling, as well as the data ratio between the vanilla sampling method and the entropy-guided sampling method, are respectively provided in Section 4.2 and Appendix C.

To further facilitate reproducibility, we provide detailed code in the Supplementary Material, including the entropy-guided sampling, SFT and DPO. Upon acceptance, we will make our experimental data and model checkpoints publicly available.

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

| Method | Mathematical-Reasoning Tasks | | | | | | Knowledge-Intensive Tasks | | | | Avg. |
|---|---|---|---|---|---|---|---|---|---|---|---|
| | AIME24 | AIME25 | AMC23 | MATH | MATH500 | GSM8K | HQA | 2Wiki. | MSQ | Bamb | |
| Prompting-Based | 6.7 | 13.3 | 47.5 | 73.8 | 62.2 | 69.4 | 21.1 | 23.8 | 9.9 | 25.5 | 35.3 |
| Tool-Light (w. SFT) | 30.0 | **26.7** | 65.0 | 85.6 | 77.2 | 89.4 | 54.7 | 55.7 | 22.8 | 58.8 | 56.6 |
| Tool-Light (w. PA.) | 30.0 | **26.7** | 67.5 | 85.8 | **80.4** | 90.8 | 55.5 | **56.5** | 24.3 | 56.0 | 57.4 |
| Tool-Light (w. SE.) | | | | | | | | | | | |
| *w. SE. 1 Loop* | 30.0 | **26.7** | 67.5 | 86.4 | 78.2 | **92.2** | 56.7 | 55.6 | **26.4** | **59.5** | 57.9 |
| *w. SE. 2 Loop* | **33.3** | 23.3 | 67.5 | **87.4** | 79.0 | 92.0 | **57.7** | 56.1 | 25.0 | 58.7 | **58.0** |

Table 3: Full results of Tool-Light on 10 reasoning tasks, with the best two results highlighted in **bold** and underline. We use Qwen2.5-7B-Instruct as the backbone model. Abbreviation: 2Wiki. (2WikiMultiHopQA). HQA. (HotpotQA). MSQ. (MuSiQue). Bamb. (Bamboogle). SE. (only conducting pre-alignment DPO training after SFT). PA. (conducting pre-alignment training and several self-evolved DPO alignment loops after SFT).

# A    THE USE OF LARGE LANGUAGE MODELS (LLMS)

In this study, we use LLM to assist in literature reading, and we do not use any large language models during other research ideation and/or writing phase.

# B    MORE DETAILS OF MAIN RESULTS

**Implementation Details.**    We use LoRA training method (Hu et al., 2022). During training, the LoRA rank is set to 8, the epoch is set to 3, the learning rate is 7e-6, and the batch size is 8. For the self-evolved DPO alignment, we train for two rounds. During testing, we set the temperature to 1 and limit the maximum call count for each tool to 4. For knowledge-intensive tasks, we use E5-base-v2 (Wang et al., 2022) with Wikipedia retrieval based on FlashRAG's settings (Jin et al., 2025b). For mathematical-reasoning tasks, we use Bing Web Search for retrieval. To avoid overly lengthy during web search, we introduce Web Browser mechanism following Tool-Star.

**Full Results at Tool-Light's Different Stages**    We present the full results of Qwen2.5-7B-Instruct model at Tool-Light framework's different stages. The results can be found in Table 3. From these results, we can draw the following insights: (1) As each training phase of Tool-Light progresses, the performance of the model gradually improves. For example, the performance of the model has significantly improved after SFT. After multiple rounds of DPO training, its performance further increases. (2) As the self-evolved DPO alignment loops proceed, the model's performance gradually converges. The performance of the model with the self-evolved DPO alignment is better than with only pre-aligned DPO training. However, with the increase of self-evolution rounds, the model's performance improves slowly. This may be because it is difficult to further generate samples beneficial for model training, resulting in only a small improvement in the model's performance after training.

# C    IMPACT OF DIFFERENT DATA RATIOS

In this section, we explore different data ratios' impact of two sampling methods on training performance. We sample data using the vanilla method and the entropy-guided method separately. Then, we mix the data in ratios ranging from 1:7 to 7:1. Meanwhile, we ensure that other settings remain unchanged. The model is then retrained in the first round of DPO, and the results are shown in Figure 6.

Results show that the model's performance in the TIR task is strongly related to the mixing ratio of the two data types. As the proportion of data with the vanilla method increases, both the model's performance and the number of tool calls improve. This is likely because training on this type of data makes the model more inclined to use tools to solve problems. However, the Efficiency metric does not continuously improve with the increasing data ratio, likely due to excessive tool call reducing

**Relationship between Various Metrics and Dataset Ratio**

Figure 6: The variation of metrics with changes in the data ratio.

the efficiency of the model's reasoning. Based on the above analysis, we set the data ratio used in the experiment to 13:7.

# D  CASE STUDY

To better demonstrate the superiority of Tool-Light over other methods in efficiency and necessity metrics, we conduct a case study. We compare the reasoning trajectories of Tool-Light with those of Tool-Star. As shown in Table 4, when addressing the same problem, even if the final answer is correct, Tool-Light's result achieves fewer tool calls compared to that of Tool-Star. And as shown in Table 5, when faced with a more difficult problem, the Tool-Light method will proactively call external tools to assist in answering questions when necessary.

Table 4: A comparison example between Tool-Light and Tool-Star on MATH500 dataset.

---

**Example #1 from MATH500 Dataset**

**Question:**
The square root of $t$ is greater than 2 and less than 3.5. How many integer values of $t$ satisfy this condition?
**Golden Answer:**
8

---

**Tool-Light's TIR Stragety**

<think> To determine how many integer values of $t$ satisfy the condition $2 < \sqrt{t} < 3.5$, we can follow these steps:
1. Square all parts of the inequality to eliminate the square root:

$$2^2 < t < (3.5)^2$$

This simplifies to:
$$4 < t < 12.25$$

2. Identify the integer values of $t$ that fall within this range. The integers between 4 and 12.25 are 5, 6, 7, 8, 9, 10, 11, and 12.
3. Count the number of these integers.
Let's implement this in Python to ensure accuracy. </think><python>

```python
# Define the range
lower_bound = 4
upper_bound = 12.25

# Find the integer values in the range
integer_values = [t for t in range(int(lower_bound) + 1, int(
    upper_bound) + 1)]

# Count the number of integer values
num_integer_values = len(integer_values)
print(num_integer_values)
```

</python><result> 8 </result> <answer>The number of integer values of $t$ that satisfy the condition $2 < \sqrt{t} < 3.5$ is $\boxed{8}$.</answer>

---

**Tool-Star's TIR Stragety**

<think>Okay, so I need to figure out how many integer values of t satisfy the condition that the square root of $t$ is between 2 and 3.5. Hmm, let me start by understanding what exactly the question is asking.
First, the square root of t has to be greater than 2 and less than 3.5. That means $\sqrt{(t)} > 2$ and $\sqrt{(t)} < 3.5$. But since we're dealing with real numbers, maybe even irrational ones, but the question specifies that t must be an integer. So $t$ has to be a perfect square between 4 (since 2 squared is 4) and... well, 3.5 squared. Let me compute that.
</think><python>

```python
lower = 2 ** 2
upper = 3.5 ** 2
print(f'Lower bound: {lower}, Upper bound: {upper}')
```

</python><result> Lower bound: 4, Upper bound: 12.25 </result>So $\sqrt{(t)}$ needs to be strictly greater than 2, which gives $t > 4$, and strictly less than 3.5, which gives $t < (3.5)^2 \approx 12.25$. Therefore, t must be integers from 5 up to 12 inclusive. To confirm:
</think><python>

```python
start = int(2**2) + 1
end = int(3.5**2)
count = end - start + 1
print(f'Start: {start}, End: {end}, Count: {count}')
```

</python><result> Start: 5, End: 12, Count: 8 </result><think>Therefore, the number of integers satisfying the condition is indeed 8. </think><answer> $\boxed{8}$ </answer>

---

Table 5: A comparison example between Tool-Light and Tool-Star on AIME24 dataset.

---

**Example #2 from AIME24 Dataset**

**Question:**
Find the largest possible real part of

$$(75 + 117i)z + \frac{96 + 144i}{z}$$

where $z$ is a complex number with $|z| = 4$.
**Golden Answer:**
540

---

**Tool-Light's TIR Stragety**

<think> To solve the problem, we need to find the maximum value of the expression $(75 + 117i)z + \frac{96+144i}{z}$ where $z$ is a complex number with $|z| = 4$.

Let's represent $z$ as $4e^{i\theta}$, where $\theta$ is a real number. Then we can write:

$$(75 + 117i)z + \frac{96 + 144i}{z} = (75 + 117i)4e^{i\theta} + \frac{96 + 144i}{4e^{i\theta}}.$$

This can be simplified to:

$$4(75 + 117i)e^{i\theta} + \frac{24 + 36i}{e^{i\theta}}.$$

**(***More reasoning contentare omitted here***)**

```python
import numpy as np

# Define the complex numbers
a = 75 + 117j
b = 96 + 144j

# Define the magnitude of z
r = 4

# Initialize the maximum real part
max_real_part = -np.inf
max_theta = 0

# Evaluate the expression for a range of theta values
for theta in np.linspace(0, 2 * np.pi, 1000):
    z = r * np.exp(1j * theta)
    expr = a * z + b / z
    real_part = np.real(expr)
    if real_part > max_real_part:
        max_real_part = real_part
        max_theta = theta

print(max_real_part)
```

</python><result> 539.9979693595761 </result><answer>The largest possible real part of the expression $(75 + 117i)z + \frac{96+144i}{z}$ where $z$ is a complex number with $|z| = 4$ is $\boxed{540}$.</answer>

---

**Tool-Star's TIR Stragety**

<think> To solve the problem, we need to find the maximum value of the expression $(75 + 117i)z + \frac{96+144i}{z}$ where $z$ is a complex number with $|z| = 4$. Let's represent $z$ as $4e^{i\theta}$, where $\theta$ is a real number. This way, we can express $z$ in terms of its polar form.
**(***More reasoning contentare omitted here***)**
However, we should confirm this by calculating the critical points. The derivative of $324\cos\theta - 504\sin\theta$ is $-324\sin\theta - 504\cos\theta$. Setting this to zero gives us the critical points. But since we know the maximum value of the expression $a\cos\theta + b\sin\theta$ is $\sqrt{a^2 + b^2}$, we can directly use this formula.
Thus, the largest possible real part is:
</think><answer> $\boxed{594}$ </answer>

---

