# OpenReview forum: "Toward Effective Tool-Integrated Reasoning via Self-Evolved Preference Learning"
_ICLR.cc/2026/Conference — ICLR 2026 Poster_

### Official Review · Reviewer_nxVk · 2025-10-28

**Soundness:** 3
**Presentation:** 3
**Contribution:** 2
**Rating:** 4
**Confidence:** 4

**Summary:**

The paper studies the Tool-Integrated Reasoning for LLMs, and proposes a new framework namely Tool-Light to resolve the problem of excessive tool calls during the reasoning. The key designs of the proposed framework contain dataset construction and multi-stage fine-tuning. Specifically, for dataset construction, Tool-Light employs two sampling strategies and for multi-stage fine-tuning, it introduces a two-stage training method. To support the effectiveness of Tool-Light, extensive empirical studies are conducted.

**Strengths:**

- The authors provide extensive experiments to demonstrate the effectiveness of **Tool-Light**, particularly through the experiments in **Figure 1**, which clearly illustrate the motivation behind the work.
- The paper is clearly written and well organized, making it easy to follow.

**Weaknesses:**

- **Figure 1** lacks sufficient explanation. For instance, the meaning of *Token Index* and the specific roles of *Step 1–4* are unclear.
- In **Figure 4(c)**, it is not specified whether the response length includes the tool-calling part. The figure shows that **Tool-Light** produces shorter responses than **Tool-Star**, yet the examples in the appendix (Examples 1 and 2) suggest otherwise.
- In **Line 257**, the computational complexity is claimed to be $O(n\log m)$, but the derivation of this result is not provided.
- The authors adopt two sampling strategies but do not include an ablation study to compare their effects.
- The experiments are conducted only on the 7B model. To further validate the effectiveness of **Tool-Light**, the reviewer encourages testing on both smaller (e.g., 3B) and larger (e.g., 72B) models.
- In **Line 80**, the authors state that *Pre-Aligned DPO Training* can reduce redundant tool calls. However, it is unclear why this is the case, as *Pre-Aligned DPO Training* does not appear to include mechanisms that explicitly control the number of tool calls.
- It is recommended to report the standard deviation across multiple runs to demonstrate the robustness of the experimental results.

**Questions:**

Answer questions in Weaknesses.

---

> ### Author Response · Authors · 2025-11-21
> **Response 1/3**
>
> Thank you for your constructive comments. We are delighted that you recognized the strengths of our work! We will address your concerns one by one below:
>
> ---
>
> ### Q1: Figure 1 lacks sufficient explanation.
>
> I apologize that we do not explain the detailed meaning of Figure 1 clearly. In Figure 1, "Token Index" represents the position of a token within that step. For example, $index_i$ represents the subscript index of the $i$-th output token in a certain step. "Step" refers to the reasoning steps—from the model starting inference or receiving the result of the previous tool call, up until the next tool call is triggered or the final answer is reached—including the process of triggering the tool call (but not including the result of the tool call). Step 1-4 thus represents the process of intermediate reasoning in the first, second, third, and fourth segments. We will provide a clearer explanation in the revised version. You can find them in Figure 1.
>
> ---
>
> ### Q2: The explanation of response length is not clear, and examples in appendix are confusing.
>
> In our paper, the response length does not include the part concerning tool calls. In Figure 4c, we discuss the impact of our method on the length of the reasoning chain. However, the length of the tool call portion is largely influenced by external tools, so after comprehensive consideration, we do not include the tool call results in the length calculation. Furthermore, during DPO and SFT training, we also do not compute the loss for the tool execution result portion.
>
> In Figure 4, we provide a comparison of the output sequence length distributions for the Tool-Light and Tool-Star methods. To illustrate more clearly that the responses generated by Tool-Light are shorter than those of Tool-Star, we have compiled detailed statistics on the length of their output sequences on each benchmark:
>
> | method | aime24 | aime25 | amc23 | math | math500 | gsm8k | hotpotqa | 2wiki | bamboogle | musique | Avg. |
> | :--- | :---: | :---: | :---: | :---: | :---: | :---: | :---: | :---: | :---: | :---: | :---: |
> | Tool-Star | 2170.7 | 1577.2 | 1306.5 | 568.5 | 747.9 | 418.4 | 340.4 | 298.2 | 694.9 | 406.8 | 853.0 |
> | Tool-Light | 2107.2 | 1429.8 | 974.9 | 476.4 | 692.0 | 371.6 | 341.4 | 337.0 | 365.7 | 454.2 | 755.0 |
>
> In the revised version, we will provide an additional example to more clearly demonstrate the efficiency and conciseness of the Tool-Light model's reasoning. You can find it in Table 6.
>
> ---
>
> ### Q3: The derivation of computational complexity is not provided.
>
> I apologize that we do not provide a detailed derivation in the paper. Here, we will elaborate on the explanation of the computational complexity. Assuming we need to sample a total of $m$ trajectories, and the average trajectory length is $n$, then the time complexity for vanilla sampling is $O(mn)$. If we use the entropy-guided sampling method, we first sample a main trajectory, and then use the entropy-guided sampling method to sample the remaining $m-1$ branch trajectories. Therefore, it is similar to a tree-sampling structure, where the traversal complexity ranges between $O(n\log m)$ and $O(nm)$. However, due to factors such as the uncertain location of high-entropy nodes and the limitation of the number of sampling links by the amount of training data, it is difficult to achieve the ideal computational complexity in actual application. Nonetheless, our proposed Entropy-Guided Sampling method can still effectively reduce the inference cost of sampling training data. For example, in our main experiment, the Entropy-Guided Sampling method reduce the inference cost by about 43% compared to vanilla TIR sampling, which is sufficient to illustrate the efficiency of our designed sampling method in inference. We will discuss the computational complexity in detail in the revised version. You can find it in Section G.1 A DISCUSSION OF THE COMPUTATIONAL COMPLEXITY.

---

> ### Author Response · Authors · 2025-11-21
> **Response 2/3**
>
> ### Q4: The authors adopt two sampling strategies but do not include an ablation study to compare their effects.
>
> Due to space limitations, it is difficult for us to include the ablation study of the sampling strategy in the main text. We have placed it in Appendix C. In the ablation study in Appendix C, we adjust the ratio of training data obtained using the two sampling strategies, transitioning from 1:7 to 7:1. To conduct a more comprehensive ablation study, we additionally test the model performance when trained solely on data sampled using vanilla sampling and entropy-guided sampling, which is presented in the table below.
>
> | metrics | All entropy-guided sampling | 1:7 | 2:6 | 3:5 | 4:4 | 5:3 | 6:2 | 7:1 | All vanilla sampling |
> | :--- | :---: | :---: | :---: | :---: | :---: | :---: | :---: | :---: | :---: |
> | LLM-as-Judge | 54.3 | 53.9 | 57.5 | 57.6 | 56.9 | 60.4 | 59.6 | 60.4 | 60.5 |
> | Avg. Tool-use Counts | 1.77 | 1.83 | 1.81 | 1.84 | 2.05 | 2.06 | 2.09 | 2.13 | 2.15 |
> | Efficiency | 0.408 | 0.412 | 0.444 | 0.434 | 0.440 | 0.460 | 0.456 | 0.426 | 0.423 |
>
> In the ablation study in Appendix C, our preliminary conclusion is that: the higher the proportion of training data obtained using the Entropy-Guided Sampling method in the total training data, the more the model tends to call tools less frequently. To balance the number of tool calls and accuracy, the data obtained by the two sampling methods must be mixed. In response, we have the following conjecture: The purpose of the entropy-guided sampling we designed is to minimize inference costs while sampling TIR paths with richer content. According to recent works, aligning with formally rich training data can increase the diversity of model output. Therefore, as the proportion of training data obtained by the Entropy-Guided Sampling method increases, the output content of the model trained on it may become richer, and it may be easier for it to find reasoning methods that involve fewer tool calls and, instead, a more complex thought process, thus leading to these changes in the LJ and avg. tool-use counts metrics in Figure 6. Conversely, as the proportion of data obtained using the vanilla sampling method increases, the model's tool-calling behavior increases, causing the efficiency metric to first rise and then fall. Therefore, balancing the ratio of the two types of training data is very important.
>
> ---
>
> ### Q5: The experiments are conducted only on the 7B model
>
> In order to prove the effectiveness of the method, it is very necessary to conduct experiments on models of different sizes. However, due to the time constraints of the rebuttal, it is difficult for us to train a Tool-Light-72B model. Here, we attempt to train Tool-Light-3B and Tool-Light-14B models based on Qwen2.5-3B-Instruct and Qwen2.5-14B-Instruct, and present the results. Since the necessity metric requires comparison with other baseline results of the same model size, and most baselines do not have 3B and 14B model versions, we only report the performance and efficiency metrics here.
>
> | Model | aime24 LJ | aime24 Effi | aime25 LJ | aime25 Effi | amc23 LJ | amc23 Effi | math LJ | math Effi | math500 LJ | math500 Effi | gsm8k LJ | gsm8k Effi |
> | :--- | :---: | :---: | :---: | :---: | :---: | :---: | :---: | :---: | :---: | :---: | :---: | :---: |
> | 3B | 3.3 | 0.03 | 3.3 | 0.03 | 25 | 0.25 | 36.2 | 0.33 | 30.6 | 0.29 | 29.2 | 0.252 |
> | 3B-SFT | 16.7 | 0.15 | 13.3 | 0.117 | 47.5 | 0.45 | 80.2 | 0.794 | 70.1 | 0.69 | 84.8 | 0.842 |
> | 3B-DPO | 20 | 0.2 | 13.3 | 0.117 | 50 | 0.46 | 81 | 0.8 | 68.6 | 0.67 | 82.2 | 0.817 |
>
> | Model | hotpotqa F1 | hotpotqa Effi | 2wiki F1 | 2wiki Effi | bamboogle F1 | bamboogle Effi | musique F1 | musique Effi |
> | :--- | :---: | :---: | :---: | :---: | :---: | :---: | :---: | :---: |
> | 3B | 3.9 | 0.16 | 6.4 | 0.069 | 2.4 | 0.105 | 3.7 | 0.043 |
> | 3B-SFT | 43.5 | 0.258 | 50.5 | 0.267 | 44.8 | 0.249 | 19.0 | 0.101 |
> | 3b-DPO | 50.5 | 0.287 | 52.6 | 0.275 | 49.2 | 0.252 | 22.0 | 0.109 |
>
> | Model | aime24 LJ | aime24 Effi | aime25 LJ | aime25 Effi | amc23 LJ | amc23 Effi | math LJ | math Effi | math500 LJ | math500 Effi | gsm8k LJ | gsm8k Effi |
> | :--- | :---: | :---: | :---: | :---: | :---: | :---: | :---: | :--: | :---: | :--: | :---: | :---: |
> | 14B | 10 | 0.1 | 6.7 | 0.05 | 50 | 0.4208 | 84.4 | 0.7913 | 73.8 | 0.6632 | 91.6 | 0.8612 |
> | 14B-SFT | 33.3 | 0.2722 | 16.7 | 0.1333 | 57.5 | 0.4313 | 89.2 | 0.8087 | 81.8 | 0.6859 | 92.8 | 0.8568 |
> | 14B-DPO | 36.7 | 0.3289 | 20.0 | 0.1833 | 67.5 | 0.6267 | 89.4 | 0.8246 | 82.4 | 0.7636 | 92.6 | 0.9029 |
>
> | Model | hotpotqa F1 | hotpotqa Effi | 2wiki F1 | 2wiki Effi | bamboogle F1 | bamboogle Effi | musique F1 | musique Effi |
> | :--- | :---: | :---: | :--: | :---: | :---: | :---: | :--: | :---: |
> | 14B | 33.4 | 0.2157 | 40.9 | 0.1796 | 41.4 | 0.2622 | 17.4 | 0.0956 |
> | 14B-SFT | 49.4 | 0.1889 | 64.2 | 0.2078 | 51.3 | 0.1823 | 25.4 | 0.0861 |
> | 14B-DPO | 51.6 | 0.2242 | 69.8 | 0.2792 | 56.4 | 0.2746 | 28.3 | 0.1272 |

---

> ### Author Response · Authors · 2025-11-21
> **Response 3/3**
>
> From the experimental results, we can find that even if the model size changes, training with the toollight framework still leads to a significant improvement in both performance and efficiency metrics compared to the original IT model. This further illustrates the effectiveness and broad applicability of the toollight training framework. To fully explore the TIR efficiency task, we will train larger models in the future, hoping to better promote community development.
>
> At the same time, in order to better connect with some previous key TIR works, we will add references to works such as ToolPrefer-LLaMA, ToolLLM, Toolformer, and Chameleon in the revised version to enhance the completeness of the Tool-Light work.
>
> ---
>
> ### Q6: The description of how the Pre-Aligned DPO Training stage can reduce tool calls is not clear.
>
> In the Pre-Aligned DPO Training section of the 4.2 TWO-STAGE TIR TRAINING PARADIGM section of the paper, we detail the criteria for screening positive and negative sample pairs for the first stage of DPO training:
>
> For the same question, we select the trajectory that is correct in the final answer, correct in format, and has the lowest number of tool calls and overall information entropy as the positive example, and select the incorrect trajectory with a higher number of tool calls than the positive example as the negative example.
> We believe that this sampling criterion can effectively filter out positive and negative sample pairs with large differences in the number of tool calls, thereby teaching the model to reduce redundant tool calls during reasoning.
>
> ### Q7: It is recommended to report the standard deviation across multiple runs.
>
> Your suggestion is very sensible. Here, we use the trained model to retest all benchmarks, repeating the test 10 times independently for each dataset, and then calculate the standard deviation for each metric. The standard deviations are now reported as follows:
>
> | std | aime24 | aime25 | amc23 | math | math500 | gsm8k | hotpotqa | 2wiki |
> | :--- | :---: | :---: | :---: | :---: | :---: | :---: | :---: | :---: |
> | Lj or F1 score | 0.0423 | 0.0564 | 0.0618 | 0.0073 | 0.0146 | 0.0107 | 0.014 | 0.0184 |
> | Efficiency | 0.0518 | 0.0502 | 0.0612 | 0.0052 | 0.015 | 0.0104 | 0.0121 | 0.0118 |
> | Necessity | 0.0667 | 0.0519 | 0.0728 | 0.004 | 0.0201 | 0.0087 | 0.0165 | 0.0189 |
>
> From this result, it can be seen that the performance fluctuation of our trained model is relatively small across all three metrics, which indicates that our model performance is very robust.
>
> ---
>
> ### Reference
>
> [1] Matsutani et al. RL Squeezes, SFT Expands: A Comparative Study of Reasoning LLMs https://arxiv.org/pdf/2509.21128
>
> [2] Yuan et al. Scaling Relationship on Learning Mathematical Reasoning with Large Language Models https://arxiv.org/pdf/2308.01825
>
> [3] Pang et al. Reasoning Pattern Matters: Learning to Reason without Human Rationales https://arxiv.org/pdf/2510.12643
>
> [4] Gao et al. A Survey of Self-Evolving Agents: On Path to Artificial Super Intelligence https://arxiv.org/pdf/2507.21046
>
> [5] Shinn et al. Reflexion: Language Agents with Verbal Reinforcement Learning https://dl.acm.org/doi/10.5555/3666122.3666499
>
> [6] Chen et al. Advancing Tool-Augmented Large Language Models: Integrating Insights from Errors in Inference Trees https://arxiv.org/pdf/2406.07115
>
> [7] Qin et al. ToolLLM: Facilitating Large Language Models to Master 16000+ Real-world APIs https://arxiv.org/pdf/2307.16789
>
> [8] Schick et al. Toolformer: Language Models Can Teach Themselves to Use Tools https://arxiv.org/pdf/2302.04761
>
> [9] Lu et al. Chameleon: Plug-and-Play Compositional Reasoning with Large Language Models https://arxiv.org/pdf/2304.09842

---

### Official Review · Reviewer_mPkz · 2025-10-31

**Soundness:** 3
**Presentation:** 3
**Contribution:** 3
**Rating:** 6
**Confidence:** 5

**Summary:**

The paper studies “tool-intelligence” in LLM agents and targets two failure modes: overthinking (too many tool calls/long chains) and underuse (skipping necessary tools). It first runs a pre-experiment that tracks token-level entropy over reasoning traces and observes patterns around tool-call moments. Building on this, it proposes Tool-Light, a two-stage pipeline: (1) SFT to establish a reasonable tool-using policy, then (2) self-evolved DPO using curated preference pairs to favor accurate, tool-efficient traces. In inference, it adds entropy-guided branching that expands alternatives at high-uncertainty steps while keeping compute controlled. Experiments span math and knowledge-intensive QA with web search and a code interpreter, reporting accuracy alongside two bespoke metrics—Efficiency (tool economy) and Necessity (using tools when needed). Results show competitive or better accuracy with fewer tool calls and shorter outputs compared to strong baselines (e.g., Tool-Star, Search-R1), plus initial ablations on sampling knobs. The paper positions entropy as a practical signal for both training supervision and test-time exploration control.

**Strengths:**

(1) Actionable diagnostic. Using token-level entropy to profile traces is simple, implementation-light, and yields intuitive visualizations to reason about when/why tools are called.

(2) Modular training recipe. The SFT → self-evolved DPO pipeline is straightforward to adopt on top of popular backbones and existing tool-use frameworks; no exotic infra required.

(3) Competitive results. Shows strong performance relative to recognized baselines while reducing tool calls—evidence that “lighter” tool use need not sacrifice accuracy.

**Weaknesses:**

**Weaknesses**

(1) Missing details. The pre-experiment omits decoding settings. No temperature, logit scaling, or sampling vs. greedy are reported. These choices change entropy levels and trends. Please specify the exact decoding config used to measure entropy and justify it. You say the entropy study spans “multiple QA datasets,” but the figure does not list them. It’s also unclear whether the same model/datasets are used later to train Tool-Light. Please enumerate the datasets in Figure 1 and clarify any reuse. For the method, entropy decides branching at top-k steps and uses top-i prefix averages. The values of k and i and their stability are not reported.

(2) Missing ablations. Table 2 varies loop count and a few sampling knobs only. Missing: (i) no-entropy (vanilla) baseline, (ii) entropy-only,  (iii) β sensitivity in DPO, (iv) reference-policy choice, and (v) branch-width sensitivity.

(3) “Multi-tool” over-claim. Most experiments use only two tools (web search + code interpreter), and several evaluations appear to be single-tool setups. There’s no test on unseen tool types or cross-tool composition tasks. As a result, the paper does not yet support broad “multi-tool generalization” claims.

**Suggestions**

(1) Control for length. Lower entropy often comes with shorter or more templated outputs. Your own results show Tool-Light reduces sequence length (Fig. 4). The same factor could explain both “lower entropy” and “fewer tool calls.” Please report entropy at fixed token positions or use length-normalized entropy.

(2) Link entropy to correctness. “Low-entropy chains use fewer tool calls” may reflect early commitment, not better answers. Please test whether entropy predicts correctness. Report AUROC for path-level mean or area-under-entropy, controlling for tool-call count.

(3) Stress test with noisy tools. Overthinking and underuse show up when tools are imperfect. Add controlled corruptions: vary retrieval precision/recall, inject code stderr/noise. Show how Tool-Light adapts tool frequency and preserves accuracy vs. Tool-Star/Search-R1. This directly probes the “analysis paralysis” claim.

**Questions:**

(1) Entropy dips after tool calls.
Could the “drop before the next tool call” be driven by inserting long, deterministic tool outputs (i.e., context length/format effects) rather than better reasoning? What happens if you replace tool results with semantically equivalent, length-matched paraphrases? Does the pattern remain?

(2) Scope to multi-tool.
Do the pre-experiment findings hold in true multi-tool settings, not just single-tool or search-heavy cases? Please show the same entropy analysis when multiple heterogeneous tools are available.

---

> ### Author Response · Authors · 2025-11-21
> **Response 1/3**
>
> Thank you for your constructive comments. We are delighted that you recognized the strengths of our work! We will address your concerns one by one below:
>
> ---
>
> ### Q1: Some experiment settings are missing.
> - The pre-experiment omits decoding settings.
>
> We apologize for the omission. The decoding parameters were set as follows: Temperature=0.6, Top-$p$=0.95, Top-$k$=20, Min-$p$=0, and Repetition Penalty=1.1. We select a temperature of 0.6 to balance the trade-off between diversity and coherence: it ensures sufficient entropy variation between trajectories while preventing excessive randomness associated with higher temperatures.
>
> - Preliminary experiment does not list datasets used in preliminary experiment.
>
> Preliminary experiment uses 2WikiMultiHopQA and Musique, using the same test splits as the main results (Section 5.2). For clarity, unless otherwise noted, our full experiment suite includes:
> 1. Mathematical-Reasoning Tasks: AIME24, AIME25, AMC23, MATH, MATH500, GSM8K.
> 2. Knowledge-Intensive Tasks: HotpotQA, 2WikiMultiHopQA, Musique, Bamboogle.
>
> - The parameters of k and i and their stability are not reported.
>
> We set top-$k=3$, selecting the three branches with the highest entropy. With a branching factor of 3, this yields a total of 10 trajectories per question (1 main chain + 9 branches). Branching positions are determined by the maximum average token-level entropy calculated over the first 10–50 tokens. To demonstrate parameter stability, we varied top-$k$ across representative datasets (AIME24, MATH, GSM8K for Math; 2Wiki, Bamboogle, Musique for Knowledge). The results are shown below:
>
> | Model | aime24 LJ | aime24 Effi | math LJ | math Effi | gsm8k LJ | gsm8k Effi | 2wiki F1 | 2wiki Effi | bamboogle F1 | bamboogle Effi | musique F1 | musique Effi |
> | :--- | :---: | :---: | :---: | :---: | :---: | :---: | :---: | :---: | :---: | :---: | :---: | :---: |
> | top-1 | 26.7 | 0.333 | 85.4 | 0.8144 | 92.2 | 0.9103 | 57.1 | 0.2398 | 58.5 | 0.2773 | 25.4 | 0.0903 |
> | top-3 | 33.3 | 0.317 | 87.4 | 0.8275 | 92 | 0.9077 | 56.1 | 0.2457 | 58.7 | 0.2858 | 25 | 0.1036 |
> | top-5 | 26.7 | 0.249 | 86.6 | 0.8209 | 89.8 | 0.8852 | 58.7 | 0.2383 | 51.2 | 0.2093 | 25.8 | 0.0926 |
> | top-7 | 26.7 | 0.238 | 84.4 | 0.8069 | 90 | 0.8848 | 59 | 0.2405 | 49.6 | 0.1886 | 23.9 | 0.0861 |
>
> While increasing top-$k$ (adding more branching points) can theoretically dilute trajectory diversity and slightly reduce performance on specific datasets (e.g., AIME24), our method generally demonstrates robustness to $k$ variations. We attribute this stability to our sampling criterion, which effectively filters high-quality positive/negative pairs regardless of the precise top-$k$ value.
> Regarding top-$i$, since it similarly influences branching point selection, we expect comparable stability. We will include a full sensitivity analysis for top-$i$ in the revised revision.
>
> ---
>
> ### Q2: Some ablation experiments are missing.
>
> We include the full ablation study of the ratio of entropy-sampled data in Appendix IMPACT OF DIFFERENT DATA RATIOS, evaluating ratios of entropy-guided to vanilla sampling data ranging from 1:7 to 7:1. We also introduce "All Entropy" and "All Vanilla" baselines for completeness. Results are summarized below:
>
> | metrics | All entropy-guided sampling | 1:7 | 2:6 | 3:5 | 4:4 | 5:3 | 6:2 | 7:1 | All vanilla sampling |
> | :--- | :---: | :---: | :---: | :---: | :---: | :---: | :---: | :---: | :---: |
> | LLM-as-Judge | 54.3 | 53.9 | 57.5 | 57.6 | 56.9 | 60.4 | 59.6 | 60.4 | 60.5 |
> | Avg. Tool-use Counts | 1.77 | 1.83 | 1.81 | 1.84 | 2.05 | 2.06 | 2.09 | 2.13 | 2.15 |
> | Efficiency | 0.408 | 0.412 | 0.444 | 0.434 | 0.440 | 0.460 | 0.456 | 0.426 | 0.423 |
>
> The results demonstrate that incorporating entropy-guided sampling effectively reduces redundancy. As the proportion of entropy-guided data increases, the model learns to solve problems with fewer tool calls (lower Avg. Tool-use Counts), likely by discovering diverse yet efficient reasoning paths. A balanced ratio (e.g., 5:3 or 6:2) achieves optimal efficiency without compromising the final performance (LLM-as-Judge).
>
> Besides, our experiments indicate that the method is robust to variations in the DPO parameter $\beta$ and the choice of reference policy. We observed no significant performance fluctuations within reasonable ranges. Detailed sensitivity plots will be included in the revised revision for completeness.
>
> As discussed before, when the total trajectory count is fixed, branch width is inversely related to the number of branching points (top-$k$). Given the stability observed in the top-$k$ ablation, we infer similar robustness regarding branch width. We contend that the data sampling criteria and the self-evolution paradigm are the primary drivers of performance, rather than specific hyper-parameters like branch width. We will include a formal confirmation of this in the revised version.

---

> ### Author Response · Authors · 2025-11-21
> **Response 2/3**
>
> ### Q3: "Multi-tool" method is over-claimed though there are two main tools.
>
> There are inaccuracies in our phrasing in the introduction section of the paper. Our work primarily focuses on mathematical reasoning tasks and knowledge-intensive tasks, both of which test the model's information retrieval and computational-planning capabilities. Search tools and code interpreters can provide excellent assistance for the model to complete these tasks. In the experimental setup of this paper, introducing more tools does not provide additional help for the model to complete these two types of tasks. We are very willing to revise the description regarding multi-tool usage in the introduction section. You can find them in Section Introduction. In future work, we will also continue to explore the field of general TIR tasks to verify the applicability and effectiveness of the Tool-Light framework.
>
> ---
>
> ### Q4: Further exploration of length control with the aid of length-normalized entropy.
>
>
> Thank you for your suggestion! We investigate the length-normalized entropy of the Tool-Light model. We select the untrained Qwen2.5-7B-Instruct (it) model, the SFT version of the Tool-Light model, and the Self-Evolved DPO version of the Tool-Light model, re-run inference on all benchmarks, and record the average output token length and the value of the length-normalized entropy. We define the calculation formula for length-normalized entropy as:
> $Length-normalized\ Entropy=\frac{1}{n}\sum_{i=1}^n\frac{Entropy_i}{length_i}$. The results are shown as follows:
>
> | Method | Avg. Length (Token) | Length-Normalized Entropy |
> | :--- | :---: | :---: |
> | Qwen2.5-7B-it | 71 | 0.3118 |
> | ToolLight-7B-SFT | 212.8 | 0.4223 |
> | ToolLight-7B | 139.3 | 0.2476 |
>
> We can draw the following conclusions:
> - SFT Increases Complexity: The SFT model exhibits longer average lengths (212.8) compared to the base model (71.0), reflecting the acquisition of complex reasoning and tool-use capabilities.
> - Tool-Light Optimizes Efficiency: Crucially, the full Tool-Light pipeline significantly reduces both sequence length (139.3) and normalized entropy (0.2476) compared to the SFT version. This confirms that our method does not simply truncate output; rather, it actively eliminates redundant reasoning steps and increases model certainty.
> - Theoretical Alignment: These findings align with established research indicating that RL-based preference optimization (including DPO and GRPO) tends to sharpen the output distribution. The reduction in normalized entropy corroborates that Tool-Light successfully internalizes the "efficient TIR" paradigm driven by our low-entropy data filtering and DPO training.
>
> ---
>
> ### Q5: Test whether entropy predicts correctness.
>
> The question you raised is very crucial. We re-select the aime24, aime25, amc23, 2wiki, musique, and bamboogle datasets, use the model to perform inference on them, and record the number of tool calls, entropy value, and correctness of the answer for each trajectory. Then, we calculate their AUROC metric, which results in the following:
>
> | Math - Tool Counts | Math - AUROC | Knowledge - Tool Counts | Knowledge - AUROC |
> | :--- | :---: | :--- | :---: |
> | 0 | 0.4732 | 0 | None |
> | 1 | 0.7772 | 1 | 0.625 |
> | 2 | 0.6667 | 2 | 0.5546 |
> | 3 | 0.5268 | 3 | 0.6035 |
> | 4 | None | 4 | 0.683 |
> | 5 | None | 5 | 0.3571 |
> | avg | 0.7793 | avg | 0.5802 |
>
> This experiment preliminarily verifies the view of our pre-experiment:
>
> For mathematical reasoning tasks, the average AUROC is $0.7793$, which indicates that on this task, predicting the correctness of the answer based on the model's output entropy distribution is relatively reasonable. For knowledge-intensive tasks, its AUROC is also higher than $0.5$. This shows that the distribution of the model's output entropy also has a certain correlation with the correctness of the answer in this task, although the effect is not as strong as in mathematical reasoning.
> When the number of tool calls is 0 in mathematical reasoning tasks, the model's confidence is instead wrong. This suggests that the untrained model may suffer from the problem of "not calling a tool when it is needed."
> When the number of tool calls is too high in knowledge-intensive tasks, the model's calibration ability may collapse. This may be because the existing reasoning chain is too long (too many tool calls), leading to reasoning failure in the model, thereby decoupling the model's confidence from its correctness.

---

> ### Author Response · Authors · 2025-11-21
> **Response 3/3**
>
> ### Q6: Stress test when receiving noisy tool results.
>
> To evaluate the robustness of our model against imperfect tool execution, we designed a comprehensive noise injection protocol:
>
> - Retrieval Noise: For search engine calls, we randomly replace 50% documents with irrelevant content to simulate retrieval degradation.
>
> - Execution Noise: For code interpreter calls, we stochastically inject syntactic errors and semantic perturbations (e.g., illegal tokens, unpaired quotes, variable renaming, numerical modification) into the feedback.
>
> We evaluated performance on MATH, MATH500, 2WikiMultiHopQA, and Musique:
>
> | Model | math LJ | math Effi | math500 LJ | math500 Effi | 2wiki F1 | 2wiki Effi | musique F1 | musique Effi |
> | :--- | :---: | :---: | :---: | :---: | :---: | :---: | :---: | :---: |
> | Qwen2.5-7B-it | 80 | 0.7175 | 62.2 | 0.4893 | 23.8 | 0.1446 | 9.9 | 0.0627 |
> | w. noisy | 80.6 | 0.4339 | 61.8 | 0.3238 | 14.3 | 0.0849 | 7.5 | 0.0424 |
> | Tool-Star | 85.6 | 0.7546 | 76.6 | 0.5226 | 53.5 | 0.2359 | 21.9 | 0.0931 |
> | w. noisy | 83.6 | 0.5609 | 76.2 | 0.5338 | 51.0 | 0.2334 | 21 | 0.0914 |
> | Tool-Light | 87.4 | 0.8275 | 79 | 0.7542 | 56.1 | 0.2457 | 25 | 0.1036 |
> | w. noisy | 86.4 | 0.548 | 74.4 | 0.4933 | 57.5 | 0.2396 | 26.5 | 0.0972 |
>
> From the experimental results, we can find:
> - While noise universally degrades performance, Tool-Light exhibits superior resilience. Remarkably, our Noisy Tool-Light (e.g., MATH 86.4) still outperforms Clean Baselines (Qwen-Instruct 80.0, Tool-Star 85.6).
> - Efficiency metrics naturally drop as models expend extra steps on error recovery, yet Tool-Light maintains the highest relative efficiency among all methods.
>
> To more intuitively show our model's resistance to interference from noisy tool calls, we will provide an example in the revised version. You can find it in Section F.1 STRESS TEST WITH NOISY TOOLS.
>
> ---
>
> ### Q7: The characteristics of the entropy pattern after obtaining semantically equivalent paraphrases.
>
> To verify that the observed entropy reduction is intrinsic to the reasoning process rather than an artifact of specific retrieval patterns (e.g., verbatim copying), we conduct a paraphrasing control experiment on knowledge-intensive tasks.
>
> Method: We employ Qwen2.5-32B-Instruct to rewrite all retrieval results, altering their lexical representation while strictly preserving semantics (Temperature=0.6). This decouples the semantic information from the specific surface form of the tool output. You can find the result in Section F.2 ENTROPY DIPS AFTER TOOL CALLS.
>
> Findings: Even with paraphrased inputs, the trajectory entropy consistently exhibits the same "decreasing trend" prior to the next tool call. This confirms that the entropy dip reflects the model's increasing internal certainty as it digests information, independent of the specific text format of the tool feedback.
>
> ---
>
> ### Q8: The exploration of entropy in TIR is limited to a single tool only.
>
> To demonstrate generalizability, we extended our evaluation to heterogeneous tool settings (e.g., Code Interpreters and Search):
>
> 1. **Mathematical Reasoning with Code Interpreters** Despite the sparser tool usage and longer CoT reasoning typical of math tasks, we observed a consistent correlation between lower entropy and fewer tool calls, validating our efficiency hypothesis in code-generation contexts.
>
> 2. **Heterogeneous Tools (GPQA Benchmark)** In complex tasks requiring alternating retrieval and computation, the discriminative power of entropy remains robust. While the specific entropy curve varies, low-entropy trajectories consistently map to efficient paths with minimal redundancy.
>
> These results confirm that our entropy-guided sampling strategy is tool-agnostic. It effectively identifies efficient reasoning patterns regardless of whether the task involves homogeneous (search-only) or heterogeneous (search + code) tool usage. Detailed plots and data are included in the revised revision. You can find them in Section F.3 SCOPE TO MULTI-TOOL.
>
> ---
>
> ### Reference
>
> [1] Matsutani et al. RL Squeezes, SFT Expands: A Comparative Study of Reasoning LLMs https://arxiv.org/pdf/2509.21128
>
> [2] Yuan et al. Scaling Relationship on Learning Mathematical Reasoning with Large Language Models https://arxiv.org/pdf/2308.01825
>
> [3] Pang et al. Reasoning Pattern Matters: Learning to Reason without Human Rationales https://arxiv.org/pdf/2510.12643
>
> [4] Cui et al. The Entropy Mechanism of Reinforcement Learning for Reasoning Language Models https://arxiv.org/pdf/2505.22617
>
> [5] Wu et al. It Takes Two: Your GRPO Is Secretly DPO https://arxiv.org/pdf/2510.00977
>
> [6] Shao et al. DeepSeekMath: Pushing the Limits of Mathematical Reasoning in Open Language Models https://arxiv.org/pdf/2402.03300
>
> [7] Liu et al. Lost in the Middle: How Language Models Use Long Contexts https://arxiv.org/pdf/2307.03172
>
> [8] Shi et al. Look Back to Reason Forward: Revisitable Memory for Long-Context LLM Agents https://arxiv.org/pdf/2509.23040

---

### Official Review · Reviewer_WYvp · 2025-11-01

**Soundness:** 3
**Presentation:** 3
**Contribution:** 3
**Rating:** 8
**Confidence:** 4

**Summary:**

This work addresses inefficiencies in Tool-Integrated Reasoning (TIR), where LLMs overuse, underuse, or mismanage tool calls. The authors analyze tool-usage dynamics through information entropy, finding that tool call results significantly influence reasoning entropy patterns. Building on this insight, they propose Tool-Light, a framework for more efficient and accurate TIR. Tool-Light combines self-evolved dataset construction—using vanilla and entropy-guided sampling with strict positive-negative selection—with a two-stage training scheme of Supervised Fine-Tuning and Self-Evolved DPO. Experiments on 10 datasets (e.g., AIME24, MATH, HotpotQA, etc) show that Tool-Light notably enhances both the efficiency and accuracy of tool-integrated reasoning.

**Strengths:**

1. Tool-Integrated /Agentical reasoning is a important topic with many potential practical applications.
2. The information-entropy based analysing after calling tools is novel and insightful.
3. The experiment evaluation is comprehensive and the performance boost is obvious.

**Weaknesses:**

1. Discuss more related work from the Self-Evolved Preference Learning perspective, such as Zeng et al., Evolving LLMs' Self-Refinement Capability via Synergistic Training-Inference Optimization, and Su et al., Trust Region Preference Approximation: A Simple and Stable Reinforcement Learning Algorithm for LLM Reasoning.
2. Clarify whether the information entropy observation generalizes across all tools, and list the specific tools used in the experiments for completeness.

**Questions:**

Please see the above Weaknesses.

---

> ### Author Response · Authors · 2025-11-21
>
> Thank you for your constructive comments. We are delighted that you recognized the strengths of our work! We will address your concerns one by one below:
>
> ---
>
> ### Q1: Discuss more related work from the Self-Evolved Preference Learning perspective
>
> Your suggestion is absolutely correct! We will add citations to these two papers in the revised version. You can find the places for modification in Section RELATED WORK. These two papers first revealed the essential lack of self-correction capability in LLMs, laying the true theoretical and experimental foundation for Self-Refinement research; simultaneously, by combining preference optimization with the trust region idea, they stand as one of the optimal paradigms for preference learning in reasoning tasks! :)
>
> ---
>
> ### Q2: Clarify the entropy observation across all tools, and list the specific tools used in the experiments.
>
> Your proposal is quite correct. In the experiments, the core tools we use are the search tool (including a local-based retriever and a Bing Search-based search engine) and the code interpreter (which accepts a piece of code and returns the compilation result).
>
> To fully explore the trend of entropy change in the model's TIR trajectories in the entity question answering task, we additionally introduce a code compiler to conduct experiments on three Mathematical-Reasoning Tasks in preliminary experiments: AIME24, AIME25, and AMC23, and record the information entropy of each step of the model's output trajectory. We will include the experimental results in the revised version. You can find them in Section SUPPLEMENTARY CONTENT FOR WYVP. Analyzing these results, we can draw the following conclusions:
>
>   1. Even for Mathematical-Reasoning Tasks that involve a code interpreter, the trajectory entropy distribution still exhibits the characteristic that "low-entropy chains tend to involve fewer tool calls,".
>   2. We also find that, compared to Knowledge-Intensive Tasks, the model's tool-calling frequency is not very high when performing Mathematical-Reasoning Tasks (solving most problems only requires calling the tool once), and the model's thought process is also longer compared to Knowledge-Intensive Tasks.
>
> ---
>
> ### Reference
>
> [1] Zeng et al. Evolving LLMs' Self-Refinement Capability via Synergistic Training-Inference Optimization https://arxiv.org/pdf/2502.05605
>
> [2] Su et al. Trust Region Preference Approximation: A simple and stable reinforcement learning algorithm for LLM reasoning https://arxiv.org/pdf/2504.04524

---

### Official Review · Reviewer_2Xcg · 2025-11-01

**Soundness:** 3
**Presentation:** 3
**Contribution:** 3
**Rating:** 6
**Confidence:** 3

**Summary:**

This paper presents Tool-Light, a framework designed to enhance the efficiency of tool-integrated reasoning (TIR) in large language models. The core empirical finding demonstrates that invoking external tools causes significant shifts in downstream token-level uncertainty, as measured by entropy. Based on this observation, the authors propose: (i) an entropy-guided sampling procedure for constructing training data, and (ii) a two-stage self-evolved Direct Preference Optimization (DPO) pipeline comprising Pre-Aligned and On-Policy phases. Experimental evaluation across ten benchmarks covering mathematical reasoning and multi-hop question answering shows comparable or superior accuracy while reducing redundant reasoning and improving both efficiency and necessity of tool usage.

**Strengths:**

1. The paper is well-structured with clear writing that facilitates comprehension of the proposed methodology.
2. The investigation of information entropy changes during tool invocation processes is particularly interesting. The authors effectively leverage these insights to guide their methodological design, providing a principled foundation for their approach.
3. While some components of the proposed data construction and self-evolved DPO framework draw upon established techniques, the overall approach remains intuitive and theoretically sound.
4. The authors conduct thorough experiments demonstrating the effectiveness of their method. The entropy distribution analysis particularly convincingly shows that the approach achieves lower entropy distributions, validating the theoretical motivation.

**Weaknesses:**

1. The analysis of entropy in tool invocation has been explored in prior work, and the conclusions drawn are not particularly surprising or groundbreaking.
2. The proposed techniques, including entropy-based sampling and evolved DPO, represent relatively incremental advances rather than significant methodological innovations.
3. While the core idea is interesting, the evaluation is restricted to DPO. The work would be significantly strengthened by demonstrating applicability to other preference learning methods such as GRPO or PPO, which would provide stronger evidence for the generalizability and robustness of the approach.

**Questions:**

N/A

---

> ### Author Response · Authors · 2025-11-21
>
> Thank you for your constructive comments. We are delighted that you recognized the strengths of our work! We will address your concerns one by one below:
>
> ---
>
> ### Q1: Entropy's analysis has been explored in prior work, and the conclusions are not surprising.
>
> In previous studies, there has indeed been many works analyzing the entropy of reasoning models. However, the core idea of our work is to use a novel approach to construct training data, starting from the perspective of entropy analysis. The method we propose can significantly reduce the inference cost of constructing data while fully augmenting the training data. Based on our research, we are the first to introduce the theory of entropy analysis into TIR data augmentation. Furthermore, unlike previous entropy studies—which all aimed to output high-entropy token sequences—to stabilize the output and guide the model towards the optimal reasoning path, we additionally selected low-entropy trajectories for DPO training, which is one of our innovative ideas.
>
> ---
>
> ### Q2: The proposed techniques do not represent significant methodological innovations.
>
> Firstly, compared to previous work studying the efficiency of TIR, our Tool-Light framework comprehensively improves the correctness of the model's tool-calling behavior (including both efficiency and necessity). This is the innovation at the motivation level of our work. Furthermore, our work enhances the model's performance at two levels: data augmentation and iterative training. In the data augmentation phase, we ingeniously retain trajectories of the low-entropy output paradigm and design a precise set of positive and negative sample pair selection criteria for training. This entropy-guided data filtering method and the filtering criteria are significant innovations of ours. Concurrently, in the training phase, we design different criteria for optimizing different aspects of TIR behavior correctness (for instance, we want the model to call tools efficiently, while also learning to call tools more often when necessary). This iterative training method, which gradually optimizes for different aspects, is also one of our innovations. In future research, we will further explore the method level to achieve more breakthroughs in the TIR efficiency task.
>
> ---
>
> ### Q3: The work do not utilize other preference learning methods such as GRPO or PPO.
>
> Thank you for your suggestion. The core of our designed training method lies in meticulously crafting a set of positive and negative sample pair selection criteria and then using DPO (Direct Preference Optimization) to make the model prefer the output trajectory paradigm of high-quality samples. The core of this DPO training method is to update parameters based on the differences between positive and negative sample pairs. However, methods like GRPO and PPO are reward-based policy optimization methods, which do not require high-quality trajectory data to support their implementation. Therefore, methods like GRPO and PPO have significant differences from our current DPO algorithm that integrates entropy sampling. Nevertheless, we believe that by carefully designing the reward, algorithms such as GRPO and PPO can also effectively enhance the model's ability to complete TIR tasks efficiently and accurately. Unfortunately, due to the tight rebuttal schedule, it is difficult at the moment to transition from the entropy-sampling-based DPO algorithm to an adaptation of methods like GRPO and PPO. In the future, we will continue to explore this area, combining entropy-guided methods with reward-based optimization algorithms to study the TIR efficiency task from more diverse perspectives.
>
> ---
>
> ### Reference
>
> [1] Wang et al. Beyond the 80/20 Rule: High-Entropy Minority Tokens Drive Effective Reinforcement Learning for LLM Reasoning https://arxiv.org/pdf/2506.01939
>
> [2] Cheng et al. Reasoning with Exploration: An Entropy Perspective https://arxiv.org/pdf/2506.14758
>
> [3] Zheng et al. First Return, Entropy-Eliciting Explore https://arxiv.org/pdf/2507.07017
>
> [4] Hao et al. Rethinking Entropy Interventions in RLVR: An Entropy Change Perspective https://arxiv.org/pdf/2510.10150
>
> [5] Dai et al. CDE: Curiosity-Driven Exploration for Efficient Reinforcement Learning in Large Language Models https://arxiv.org/pdf/2509.09675
>
> [6] Huang et al. Reinforced Internal-External Knowledge Synergistic Reasoning for Efficient Adaptive Search Agent https://arxiv.org/pdf/2505.07596
>
> [7] Qian et al. SMART: Self-Aware Agent for Tool Overuse Mitigation https://aclanthology.org/2025.findings-acl.239.pdf

---

### Author Response · Authors · 2025-11-21
**General Response**

We sincerely thank all the reviewers for their time and constructive feedback. We are thrilled that our core contributions have been recognized. Our Tool-Light framework is considered **interesting and novel** (2Xcg, WYvp). Our Tool-Light framework is also regarded as **effective and competitive** (2Xcg, mPkz, nxVk). Additionally, the reviewers believe that **our writing is clear** (2Xcg, nxVk).

The reviewers' main concerns focus on several key areas, which we have addressed through detailed responses and new quantitative analyses. We summarize the following points.

| Focus Area | Reviewer Concerns | Our Actions |
| :--- | :--- | :--- |
| Lack of Novelty | 2Xcg | **Action:** We carefully listed the research content of previous works [1-7] and detailed the novelty of our work compared to previous efforts. |
| Applicability to Other Methods | 2Xcg | **Action:** We detailed potential issues that may arise when adapting our method to other algorithms [8-9] and stated that we will use more RL algorithms to explore TIR efficiency tasks in the future. |
| Insufficient Discussion of Related Work | WYvp | **Action:** We promise to discuss related work more thoroughly in the revised version. |
| Insufficient Exploration of Generalizability in Preliminary Experiments | WYvp, mPkz | **Action:** (1) Following the settings of the preliminary experiments, we supplemented experimental results on Mathematical-Reasoning Tasks.(2) We conducted a deeper exploration of the preliminary experiments following the reviewers' suggestions and drew new conclusions.(3) Using the GPQA dataset as an example, we explored the differences between preliminary results and existing results when using multiple heterogeneous tools. |
| Missing Experimental Setup Details | mPkz | **Action:** We detailed the parameter settings used in the preliminary experiments and conducted an ablation study on the top-k experimental parameter. |
| Missing Ablation Studies | mPkz, nxVk | **Action:** (1) Regarding the ablation study on sampling methods, we have placed it in Appendix C with comprehensive supplementation.(2) Regarding the ablation study on DPO algorithm settings, we explained that we did not find the Tool-Light training framework to be highly sensitive to these settings in our experiments.(3) Regarding the exploration of branching width, we believe its impact is similar to changing top-k. |
| Overstated Multi-Tool Claims | mPkz | **Action:** (1) We elaborated on the usage of tools regarding the specific characteristics of the tasks.(2) We promise to restate the relevant content in the Introduction section of the revised version. |
| Lack of Specificity in Writing | mPkz, nxVk | **Action:** (1) We provided detailed explanations for parts that were not specific or clear.(2) We promise to rewrite relevant content in the revised version. |
| Missing Training Results on Other Model Sizes | nxVk | **Action:** (1) We used the Tool-Light framework to retrain Qwen2.5-3B-Instruct and Qwen2.5-14B-Instruct.(2) We promise to explore training on larger models in future continuous research. |
| Missing Standard Deviation in Results | nxVk | **Action:** We re-ran the experiments 10 times for each benchmark and recorded the standard deviation for each benchmark. |

We believe these clarifications and additional quantitative results have thoroughly addressed the reviewers' concerns.
**Our team has sincerely invested tremendous efforts into this work, and we sincerely hope that our work can contribute to research in the TIR field.**

Best regards,

The Authors

---
### Reference

[1] Wang et al. Beyond the 80/20 Rule: High-Entropy Minority Tokens Drive Effective Reinforcement Learning for LLM Reasoning https://arxiv.org/pdf/2506.01939

[2] Cheng et al. Reasoning with Exploration: An Entropy Perspective https://arxiv.org/pdf/2506.14758

[3] Zheng et al. First Return, Entropy-Eliciting Explore https://arxiv.org/pdf/2507.07017

[4] Hao et al. Rethinking Entropy Interventions in RLVR: An Entropy Change Perspective https://arxiv.org/pdf/2510.10150

[5] Dai et al. CDE: Curiosity-Driven Exploration for Efficient Reinforcement Learning in Large Language Models https://arxiv.org/pdf/2509.09675

[6] Huang et al. Reinforced Internal-External Knowledge Synergistic Reasoning for Efficient Adaptive Search Agent https://arxiv.org/pdf/2505.07596

[7] Qian et al. SMART: Self-Aware Agent for Tool Overuse Mitigation https://aclanthology.org/2025.findings-acl.239.pdf

[8] Wu et al. It Takes Two: Your GRPO Is Secretly DPO https://arxiv.org/pdf/2510.00977

[9] Shao et al. DeepSeekMath: Pushing the Limits of Mathematical Reasoning in Open Language Models https://arxiv.org/pdf/2402.03300

---

### Author Response · Authors · 2025-11-27
**A Kind Reminder for Reading the Response**

Dear Reviewers,

We have revised the paper and added many additional results to address your comments. Since the rebuttal period is closing very soon, can you please check the response to see whether it mitigates your concerns? We would greatly appreciate that!

Thank you,
The authors of Tool-Light.

---

### Meta-Review · Area_Chair_Pszs · 2026-01-07

**Summary:**

Reviewers find Tool-Light well written, empirically solid, and practically useful: entropy shifts around tool calls are leveraged to build better training data and a two-stage SFT→self-evolved DPO pipeline that improves both correctness and tool-use efficiency across many math and multi-hop QA benchmarks. The main concerns were novelty (entropy analysis and DPO feel incremental), scope/generalizability (initially DPO-only, limited tool diversity, mostly 7B), and missing clarity/ablations/cost/robustness details. The rebuttal meaningfully strengthens the paper by adding key missing experimental settings, extensive ablations and sensitivity analyses (data ratio, branching/top-k, β/reference policy), length-normalized entropy, correctness–entropy linkage (AUROC), noisy-tool stress tests, paraphrase controls, and additional model sizes (3B/14B). Overall, remaining issues are mostly “positioning” (incremental vs. breakthrough; not yet demonstrating PPO/GRPO) rather than unanswered technical gaps.

**Reviewer Concerns:**

Reviewer 2Xcg: Partly addressed. The authors argue novelty is in entropy-guided data augmentation and staged optimization, but they do not demonstrate training with GRPO/PPO (they explicitly defer it), so generality beyond DPO remains outstanding.

Reviewer WYvp: Addressed. The authors agree to add self-evolved preference-learning related work and clarify tools used (search + code interpreter), plus provide additional entropy observations with code/math tasks.

Reviewer mPkz: Largely addressed. The rebuttal adds decoding settings/dataset enumeration, broader ablations (including all-entropy/all-vanilla mixes), length-normalized entropy, entropy↔correctness AUROC, noisy-tool robustness, and paraphrase controls; the remaining limitation is that “multi-tool generalization” is still somewhat narrow (mostly search+code, with broader claims now softened).

Reviewer nxVk: Largely addressed. Authors clarify Figure 1 terminology, define response length excluding tool outputs, provide sampling-strategy ablations, add 3B/14B results, explain why pre-aligned DPO reduces tool calls via pair construction, and report multi-run standard deviations; any residual concern is mainly whether the complexity discussion is rigorous enough, but the core requested clarifications/ablations are covered.

**Reviewer Scores:**

Reviewer 2Xcg did not state a score change, and since the rebuttal does not add GRPO/PPO/GRPO-style evidence, it is most consistent to expect their score to stay the same unless they are persuaded by the added experiments and positioning.

Reviewer WYvp did not state a score change, and because their requests (related work + tool listing/generalization clarification) are directly handled, their score would likely stay the same or strengthen slightly in confidence.

Reviewer mPkz did not state a score change, and because essentially every concrete missing-detail/validation request is answered with new analyses (normalization, AUROC, noise, paraphrase, ablations), their concerns appear largely resolved, making an upward score adjustment plausible though not guaranteed.

Reviewer nxVk did not state a score change, and since the rebuttal adds the requested explanations, ablations, multi-size results, and robustness (std over 10 runs), their initial “below threshold” concerns look substantially addressed, so a score increase would be reasonable if they accept the new evidence.

---

### Decision · Program_Chairs · 2026-01-26

Accept (Poster)